# The Effectiveness of Family Medicine-Driven Interprofessional Collaboration on the Readmission Rate of Older Patients

**DOI:** 10.3390/healthcare11020269

**Published:** 2023-01-15

**Authors:** Ryuichi Ohta, Chiaki Sano

**Affiliations:** 1Community Care, Unnan City Hospital, 699-1221 96-1 Iida, Daito-cho, Unnan 699-1221, Japan; 2Department of Community Medicine Management, Faculty of Medicine, Shimane University, 89-1 Enya cho, Izumo 693-8501, Japan

**Keywords:** interprofessional collaboration, family medicine, rural, community hospital, readmission, polypharmacy, frail, nutrition

## Abstract

Interprofessional collaboration (IPC) for older patient care among family physicians, dentists, therapists, nutritionists, nurses, and pharmacists in the rural hospital care of older patients could improve the hospital readmission rate. However, there is a lack of interventional studies on IPC for improving the readmission rate among Japanese older patients in rural hospitals. This quasi-experimental study was performed on patients >65 years who were discharged from a rural community hospital. The intervention was IPC implementation with effective information sharing and comprehensive management of older patients’ conditions for effective discharge and readmission prevention; implementation started on 1 April 2021. The study lasted 2 years, from 1 April 2021 to 31 March 2022 for the intervention group and from 1 April 2020 to 31 March 2021 for the comparison group. The average participant age was 79.86 (standard deviation = 15.38) years and the proportion of men was 45.0%. The Cox hazard model revealed that IPC intervention could reduce the readmission rate after adjustment for sex, serum albumin, polypharmacy, dependent condition, and Charlson Comorbidity Index score (hazard ratio = 0.66, 95% confidence interval: 0.54–0.81). Rural IPC intervention can improve inpatient care for older patients and decrease readmission rates. Thus, for effective rural IPC interventions, family physicians in hospitals should proactively collaborate with various medical professionals to improve inpatient health outcomes.

## 1. Introduction

Interprofessional collaboration (IPC) among healthcare professionals in medical institutions is essential for improving healthcare for older patients [1,2,3]. Comprehensive care involving various professionals can improve the health of older adults by improving their oral, nutritional, physical, and cognitive conditions [1,2,3]. Considering oral care, oral bacteria overgrowth in older patients can cause mouth inflammation and direct infections [4,5], inducing systemic inflammation and causing various chronic diseases, such as diabetes, cardiovascular diseases, and osteoporosis [6,7]. Aspiration pneumonia caused by poor oral care can affect the lives of older patients [8]. Maintenance of oral hygiene by dentists and nurses is vital for preventing exacerbation and relapse of acute diseases [9].

Nutritional and rehabilitation interventions are also vital for enabling the discharge of older patients to their homes [10,11]. Older patients tend to have malnutrition owing to multimorbidity; therefore, intervention by nutritionists during admission can sustain their activities of daily living (ADL) by supporting their food intake conditions and adjusting the food form and amount for older patients [12]. Generally speaking, during admission, older patients tend to lie on their beds most of the time, which could impinge on their ADL more quickly than that of younger patients, owing to muscle loss arising from disuse syndrome [13]. Older patients’ physical and cognitive functions could be weakened more rapidly than those of younger patients [14]. The interventions of physical and occupational therapists can sustain and improve admitted older patients’ physical and cognitive functions [13].

As society ages, hospital care needs effective IPC involving various professionals, especially in the field of family medicine. Family medicine is one of the medical specialties that allows comprehensive patient care, including the medical and biopsychosocial aspects [14]. Family medicine practitioners are also termed general practitioners or general physicians. Their roles can depend on their working contexts. In aging societies, seen more in developed countries, older patients have multimorbidity and polypharmacy [15,16]. In hospitals, older patients are admitted with complications attributed to their diseases and medications, which could be treated by pharmacists [17,18]. The department of general medicine needs to accommodate many older patients in community hospitals since family medicine specializes in multimorbidity and polypharmacy [14]. In family medicine, physicians are familiar with IPC with various healthcare professionals, such as dentists, pharmacists, nutritionists, therapists, and other community-based healthcare professionals [19,20]. Collaboration between family physicians and other healthcare professionals should be driven to improve hospital care for older patients.

In Japan, society is currently aging rapidly, and the increase in the number of family physicians can improve IPC interventions in the hospital care of older patients, leading to improved clinical outcomes. In Japan, family physicians work in various settings, such as clinics and hospitals [14]. In IPC interventions in hospitals, oral care in hospitals can decrease the complications arising from poor oral hygiene, such as aspiration pneumonia and nutritional conditions [21,22]. Nutritional and physical interventions can be improved through collaboration with physicians, nutritionists, and therapists [23,24]. The improvement in IPC may result in various outcomes, including decreased readmission rates in hospitals [25,26]. In previous studies, comprehensive approaches to frail patients with various professionals improved aspiration pneumonia and mortality rates [27,28]. In Japan, family physicians are educated to collaborate effectively with various medical professionals [18]. IPC interventions for older patient care driven by family physicians can improve patient outcomes.

One of the factors that highlight the improvement in hospitalized older patient care is the decrease in readmission rate. Older patients have various chronic diseases with polypharmacy [29,30]. The nature of this multimorbidity can make older patients vulnerable to the exacerbations of chronic diseases, infections, and cardiovascular diseases [31]. Readmission of older patients increases the likelihood of morbidity and mortality, which should be prevented [32,33]. Through effective IPC of various healthcare professionals, the readmission rate can be decreased by enhancing the discharge and follow-up processes in outpatient departments.

High readmission rates increase the burden on healthcare systems. They led to increased patient numbers in hospitals, which require the commitment of various medical professionals [34]. The commitment in hospitals is typically more intensive than in that in the community [35]. The increase in hospitalized patient numbers can exhaust healthcare professionals, which may ultimately lead to burnout [36,37,38]. In addition, the cost of hospital admission is higher than community care, impinging on the financial conditions of healthcare systems [39]. Utilizing IPC intervention to decrease readmission rates of older patients can mitigate the burden on healthcare professionals and systems [40].

Considering rural Japanese areas, the medical resources could be limited, and effective IPC should be driven by the small number of medical staff. Interventions should involve various professionals and not depend on specific individuals [14]. Family physicians working in rural hospitals deal with various health issues of older patients due to a lack of medical specialties; despite this, they provide highly comprehensive care and specialize in IPC interventions with various professionals [14]. Rural family physicians as leaders should facilitate interaction with various professionals to drive collaboration in rural areas [41]. Evidence supporting IPC is still lacking in Japan, as family medicine education is a recent occurrence. A lack of evidence exists in rural Japanese areas concerning the relationship between the improvements in IPC driven by family medicine and the readmission rates of older patients. Clarification of the effectiveness of IPC driven by family physicians can help evolve family medicine education in Japan to the same level as in other countries with aging societies. Therefore, our study aimed to clarify the effect of IPC on the readmission rate of older patients to rural hospitals.

## 2. Materials and Methods

A quasi-experimental study was performed with patients aged >65 years who were admitted and discharged from a rural community hospital managed by family physicians. The intervention was the implementation of IPC with effective information sharing and comprehensive management of older patients’ conditions for effective discharge and readmission prevention; the intervention was started on 1 April 2021. The duration of the study period was 2 years, from 1 April 2020 to 31 March 2021 (comparison group), and 1 April 2021 to 31 March 2022 (intervention group). The primary outcome was the hospital readmission rate.

### 2.1. Setting

Unnan City is one of the smallest and most remote cities in Japan, and it is located southeast of an administrative unit in a rural setting. In 2020, the total population of the city was 37,638 (18,145 males and 19,492 females), and 39% of the population was over 65 years of age, which is expected to reach 50% by 2025 [42]. There are 16 clinics, 12 home care stations, 3 visiting nurse stations, and only a single public hospital in this city. At the time of the study, Unnan City Hospital had 281 care beds, including 160 acute care, 43 comprehensive care, 30 rehabilitation, and 48 chronic care beds. The nurse-to-patient ratios were 1:10, 1:13, 1:15, and 1:25 for acute, comprehensive, rehabilitation, and chronic care, respectively. The hospital had 27 physicians, one dentist, 197 nurses, 7 pharmacists, 15 clinical technicians, 37 therapists, 4 nutritionists, and 34 clerks [43].

Unnan City Hospital trains medical residents using an educational curriculum based on the Japanese Primary Care Association’s Board of Family Medicine, which was developed according to the World Standard of Education of Family Medicine [18]. During the study period, there were three medical educators who specialized in family medicine. This curriculum can be used to simultaneously educate a maximum of three residents. One resident in 2018 and 2019 and three in 2020 and 2021 were engaged in the curriculum. In 2020 and 2021, the same number of seven physicians (three family physicians and five family medicine residents) were part of the Department of Family Medicine [43].

### 2.2. Participants

The participants were patients aged >65 years discharged from Unnan City Hospital. The patients were regularly followed up at Unnan City Hospital or other medical institutions in Unnan City from 1 April 2020 to 31 March 2022. All readmissions took place at Unnan City Hospital. All the participants were followed up until readmission (range: 180–540 days).

### 2.3. IPC Intervention in the Rural Hospital

For effective IPC intervention with effective information sharing and comprehensive management of older patients’ conditions to improve inpatient care and patient support following discharge, family physicians collaborated with multiple medical professionals, including dentists, therapists, pharmacists, nutritionists, nurses, and social workers. Each collaboration was driven by family physicians with the support of medical clerks in the hospital [44]. Previously, there has been no formal IPC among healthcare professionals regarding patients in the department of family medicine in the rural hospital. On 1 April 2022, the department of family medicine started the following collaborations (Table 1). In acute conditions, dentists checked the patients’ oral condition and informed the nurses regarding the oral care approaches. Therapists checked the patients’ frailty and provided rehabilitation. Pharmacists checked the patients’ polypharmacy and adjusted the prescriptions upon discussion with the family physicians. Nutritionists checked the patients’ nutritional condition and adjusted the diet to avoid undernutrition. Nurses and social workers took care of the patients based on the assessments of the other medical professionals and assessed the social conditions to prepare for the patients’ discharge and ensure stability in their homes (Figure 1).

#### 2.3.1. Collaboration with Dentists

On 1 June 2021, Unnan City Hospital applied inpatient oral care in collaboration with the departments of family medicine, oral and maxillofacial surgery, and nursing [45]. This collaboration aimed to improve the oral care of older patients during admission. This intervention was performed in all patients aged >65 years admitted to the Department of Family Medicine.

Family medicine physicians assessed the admitted patients and discussed the need for oral care with the patients and their families. After obtaining their consent, the family physicians referred the patients to the Department of Oral and Maxillofacial Surgery; the family physicians informed the reasons for admission of the patients to the dentist and dental hygienists.

The Department of Oral and Maxillofacial Surgery consists of one dentist and two dental hygienists who assess and care for older admitted patients upon referral from the family physicians. They intensively assessed the condition of the teeth, gingiva, and mucosa of the mouth. They followed the referred patients for a week to assess any change in their oral condition. In addition, they collaborated with the nurses in each ward to improve oral care.

Collaboration between the Departments of Oral and Maxillofacial Surgery and Nursing was conducted in each ward. One nurse specializing in oral care, and a dentist and dental hygienists discussed each patient’s oral condition at the bedside [28]. The patients’ oral care instructions were revised based on their oral and disease conditions. The nurses ensured the suggested oral care approach was implemented until discharge.

#### 2.3.2. Collaboration with Physical and Occupational Therapists

On 1 April 2021, based on the hospital rehabilitation condition, such as multimorbidity impinging on the patients’ discharge and lack of information exchange between the physicians and physical and occupational therapists, the Department of Family Medicine at Unnan City Hospital collaborated with the Department of Rehabilitation. Family physicians and therapists discussed their patients from each other’s perspectives and shared ideas for the care and discharge plan of hospitalized patients [46]. The discussions were conducted once a week at an interdisciplinary conference with the family medicine and rehabilitation departments.

During the conference, different perspectives were shared. Physicians admitted the patients based on medical perspective and presented concrete symptoms and clinical findings supporting the diagnosis, treatment plan, and predicted the clinical course. Therapists shared their plans for the rehabilitation of the patients based on the physician’s diagnoses and discussed specific points and cautions in rehabilitation, such as changes in symptoms and vital signs, with the physicians.

Through these conferences, physicians and therapists share their viewpoints regarding the discharge of patients. In conferences, physicians can understand when patients achieve the levels of physical and mental condition required for their discharge based on the information on patient rehabilitation. Therapists can understand the physicians’ goals for treatment and expectations for rehabilitation, thereby facilitating appropriate rehabilitation of each patient.

#### 2.3.3. Collaboration with Pharmacists

To address the issue of polypharmacy, discussions concerning medicines of admitted patients among family physicians and pharmacists started on 1 April 2021. A discussion was conducted to enable patient prescriptions appropriate for the present medical conditions. The discussion began with the pharmacists upon admission of each patient.

Pharmacists checked the type and number of the admitted patients’ medications. They reviewed the patients’ medical histories based on the referral records of primary care doctors and previous medical records of Unnan City Hospital. The general physicians were informed when a mismatch between the patients’ medical conditions and medications was found.

The family physicians checked the prescriptions of the admitted patients based on the pharmacists’ suggestions. When the general physicians assessed the prescriptions, they discussed appropriate medications with the pharmacists based on the criteria of polypharmacy and prescribing cascade. Before patient discharge, general physicians informed the patients of the conditions for effective collaboration with the patients’ primary care physicians.

#### 2.3.4. Collaboration with Nutritionists

Collaboration with nutritionists was performed by screening the nutritional condition of the admitted patients and consultation with family physicians regarding alteration of the meals for patients upon observation of patient food intake and changes in the nutritional condition based on laboratory nutritional assessment.

The assessment of the nutritional condition of the admitted patients was performed using the mini-nutritional assessment tool, which is generally employed in Japan to assess nutritional condition using a simple questionnaire. Nutritionists consulted general physicians and provided meal plans with appropriate nutrition and followed up on the intake of meals upon assessing patients as undernourished.

The follow-up of the nutritional condition of the admitted patients was performed based on laboratory changes, such as lymphocyte counts, serum albumin, and cholinesterase levels. General physicians checked the laboratory data of patients at 3 days and 1 and 2 weeks after admission for their nutritional condition. When the laboratory markers of nutrition deteriorated, the nutritionists reviewed the nutritional condition of the patients again. Through discussions with the general physicians, nutritionists modified meals by changing the form and content.

#### 2.3.5. Collaboration with Nurses

Collaboration between physicians and nurses is vital for the effective care and management of the discharge of patients. In our hospital, we performed weekly discussions regarding patient care and discharge management in each ward among general physicians and chief nurses in addition to each patient care conference [47]. Discussions were conducted in the conference rooms of each ward.

Nurses shared issues of patient management at the Department of Family Medicine meeting every week concerning the treatment and discharge plans. Nurses were anxious about differences in the viewpoints of each physician involved in patient care in the Department of General Medicine [48]. The discussion was conducted by a general physician and several chief nurses from each ward. Participating general physicians could be informed of the current problems of general medicine patient care as a whole and individually for each patient.

The shared issues of the nurses were discussed at the department conference in the Department of Family Medicine. The issues were discussed, and revised plans were constructed and communicated to the Department of Nursing in the next discussion. In addition, the issues of each patient were shared, and each charged family physician attempted to modify their management of patients based on the suggestions of the nurses.

### 2.4. Measurements

#### 2.4.1. Primary Outcome

The primary outcome was the readmission rate to Unnan City Hospital with acute diseases, such as infections, cardiovascular disease, stroke and hemorrhage, bone fractures, and other acute symptoms. The readmission data were extracted from the electronic medical records of Unnan City Hospital. Regarding the duration of the measurement of the primary outcome, the duration of the comparison group was from 1 April 2020 to 31 August 2021, and the duration of the intervention group was from 1 April 2021 to 31 August 2022.

#### 2.4.2. Covariates

The participants’ background information was obtained from the digital health records of Unnan City Hospital. Participants’ data included age, sex, body mass index (BMI), serum albumin (g/dL) concentration for nutritional assessment, serum creatinine, estimated glomerular filtration rate for the evaluation of renal function, hemoglobin concentration, and care level based on the Japanese long-term care insurance [49]. In addition, the Charlson Comorbidity Index (CCI), based on the patients’ medical histories, including the presence of heart failure, myocardial infarction, asthma, chronic obstructive pulmonary disease (COPD), kidney disease, liver disease, diabetes mellitus, brain infarction, brain hemorrhage, hemiplegia, connective tissue disease, dementia, and cancer [50], was used to indicate the severity of their condition. The cognitive and motor components and total score of the functional independence measure at discharge were measured by the therapists. These scores served as indicators of a patient’s ADL [51].

### 2.5. Data Analysis

Data were descriptively summarized by dividing the participants into two groups. Based on the test of normality, Student’s *t*-test was performed on parametric data, and the Mann–Whitney U test was performed on non-parametric data. Based on previous studies and the averages of variables, numerical variables were dichotomized as follows: CCI (≥5 and <5) [50] and care level (≥1 and 0), based on the burden on caregivers and families [49]. The dependent condition was defined as care level ≥1. A significance level of *p* < 0.05 was used for all the comparisons. The participants were divided into two groups based on whether they received an oral intervention (intervention and control). To investigate the statistical difference in their readmission rate of 0.1, a minimum of 219 participants were required in each group based on α (alpha) = 0.05, β (beta) = 0.20, power of 80%, and mortality rate in the intervention group of 0.1 [11]. Variables reported to be significantly associated with hospital readmission in previous studies were selected and analyzed using the Cox proportional hazards regression model to determine the independent predictors of readmission following discharge. A Cox proportional hazards model adjusted for independent factors was used to calculate the hazard ratios (HR) and 95% confidence intervals (CI). Cumulative event-free survival rates were calculated using the Kaplan–Meier method and analyzed using the log-rank test. Patients with missing data were excluded from the analysis. All the statistical analyses were performed using EZR (Saitama Medical Centre, Jichi Medical University, Saitama, Japan), a graphical user interface for R (The R Foundation, Vienna, Austria) [52].

### 2.6. Ethical Consideration

The hospital ensured the anonymity and confidentiality of patient information. The study information was posted on the hospital website without disclosing any details about the patients. The contact information of the hospital representative is also listed on the website to address any questions regarding this study. All participants were informed of the purpose of the study and informed consent was obtained from all participants. The clinical ethics committee of our institution approved this study (approval code: 20210023).

## 3. Results

### 3.1. Participant Selection

Figure 2 presents the flowchart of the study population selection process. In total, 889 and 1152 patients were admitted to the Department of Family Medicine in the comparison (1 April 2020 to 31 March 2021) and intervention groups (1 April 2021 to 31 March 2022), respectively. Among them, 287 and 241 were excluded from the comparison and intervention groups, respectively, owing to the lack of acute diseases. Among the patients admitted to the Department of Family Medicine with acute diseases, 81 and 101 were excluded from the comparison and intervention groups, respectively, owing to death during hospitalization. In total, 749 and 863 participants were included in the comparison and intervention groups of this study, respectively (Figure 2).

### 3.2. Demographics of the Participants

The average age of the participants was 79.86 (standard deviation = 15.38) years and the proportion of men was 45.0%. There were significant differences in the following factors between the intervention and comparison groups: age (*p* < 0.001) and readmission rate (*p* < 0.001). (Table 2)

### 3.3. Relationship between IPC and Readmission

The Kaplan–Meier curves show the estimated probability of non-readmission as a function of the presence of IPC interventions (Figure 3). The Cox hazards model was used to investigate the relationship between IPC intervention and readmission rates. This study used the Cox hazards model with forced entry based on previous research [11,35], by including all previous independent variables. Considering the Cox hazards model, the presence of IPC intervention (HR = 0.66, 95% CI: 0.54–0.81), male sex (HR = 1.28, 95% CI: 1.04–1.58), serum albumin (HR = 1.15, 95% CI: 1.01–1.31), polypharmacy (HR = 1.56, 95% CI: 1.22–2.00), dependent condition (HR = 1.34, 95% CI: 1.03–1.76), and CCI score ≥5 (HR = 1.97, 95% CI: 1.50–2.59) were related to the readmission rates (Table 3).

## 4. Discussion

This study shows the effectiveness of rural IPC intervention driven by family medicine practitioners on the rates of readmission in rural hospitals. Readmission was related to sex, serum albumin, BMI, polypharmacy, dependent condition, and high CCI. Based on the results, rural IPC intervention involving the approaches to polypharmacy, nutritional condition, and rehabilitation in hospitals can improve interventions for discharge.

The effectiveness of rural IPC intervention driven by family medicine practitioners can be driven by comprehensive collaboration with various healthcare professionals in rural areas. Our research demonstrates that the IPC intervention involving dentists, nurses, therapists, pharmacists, and nutritionists could improve the health outcome of readmission rates in rural hospitals [26,53]. Considering rural IPC interventions, healthcare professionals need to collaborate effectively owing to the lack of healthcare resources [54,55]. Previous studies have demonstrated that IPC intervention in communities could improve the rate of discharge and readmission through collaboration with home care nurses and doctors [56,57]. These outcomes were assessed in patients with various diseases, such as heart failure and COPD [56,57]. In contrast, these studies involved few medical professionals and did not demonstrate comprehensive approaches for older patients concerning readmission in hospitals. IPC intervention requires various interactions among healthcare professionals, warranting systematic approaches that comprise interventions in the whole body, and a single-layer intervention such as two types of healthcare professionals is unable to demonstrate the effect of health outcomes in the pragmatic care of patients [24,58]. This research could thus be considered valuable since this study’s IPC intervention involves various professionals supporting patients’ lives in hospitals and their homes, even though rural areas lack healthcare resources.

This study demonstrated that male sex, high serum albumin level, low BMI, polypharmacy, dependent condition, and high CCI could be related to the readmission rates in hospitals. The relationship with males could be affected by Japanese customs toward help-seeking behaviors (HSB) [59]. Masculinity could affect their HSB since men believe that they should not show their weakness to others [60]; men tend to endure their symptoms until they become critical, especially in rural areas [61]. Compared to women, men tend to visit hospitals in advanced disease stages, often requiring emergency hospitalization and treatment.

Higher serum albumin levels could be affected by the patient’s body fluid volume at admission. In this study, the average serum albumin level was 3.7 g/dL, which is normal [62]. Patients with acute conditions may tend to lose body fluid volume [62]. Loss of body fluid volume may result in the apparent normalization of serum albumin levels [63]. Therefore, a higher serum albumin level could indicate dehydration or conditions related to volume deposition, resulting in a critical state affecting their discharge and readmission.

A low BMI could affect the readmission rate since a low BMI could be related to poor physical condition. Low BMI may be related to sarcopenia and frailty in older patients [23,64], which could be caused by not only aging but also malnutrition and a sedentary lifestyle [21,65]. Sarcopenia and frailty have been shown to increase the rates of dysphagia and mortality in hospitals [66]. The continuation of these conditions after discharge could result in a vulnerable medical state, leading to an increase in the readmission rate. In this study, rural IPC involved nutritionists and therapists who could improve the nutritional and physical conditions of sarcopenia and frailty in hospitalized patients. These interventions decreased the readmission rate in this study.

Polypharmacy and high CCI impinge on various health outcomes in older patients. This study showed that polypharmacy is related to readmission rates in rural hospitals. Previous studies have shown similar results concerning this relationship in rural settings [11,67]. Polypharmacy is associated with multimorbidity in older patients [68]. Multimorbidity tends to cause various acute symptoms and makes older patients vulnerable and at higher risk of readmission [64,69]. This study also showed that multimorbidity and comorbidities could independently cause higher readmission rates, thereby independently affecting the readmission rate. Multimorbidity could be challenging to manage owing to chronic diseases [15]. Interventions for polypharmacy can be effective. In this study, IPC with pharmacists was found to be effective in decreasing the readmission rate in rural hospitals. We did not measure the changes in the number of medicines in the participants. Future studies should investigate the relationship between the decrease in the number of medicines and the readmission rate.

Dependent conditions could affect the HSB of older adults, which could increase their readmission rates. Patients with dependent conditions may not be able to seek medical care by themselves and may require the assistance of caregivers among their families and care workers upon presenting symptoms [70,71]. Nowadays, most family members have jobs and work outside the home [72]. Older individuals with dependent conditions are unable to receive help promptly. Moreover, in the rural setting, owing to a lack of workforce, middle-aged individuals need to work in communities and are busy [41]. Therefore, older patients may be reluctant to report their symptoms, considering the burden on their families, especially in rural areas [71]. The reluctance to seek help could increase the risk of exacerbation of their symptoms and diseases, causing readmission to hospitals, as reported by previous studies [11,67].

Our study’s strength was the clarification of the effect of family medicine-driven IPC in rural areas. In this study, the family medicine department facilitated IPC in a rural hospital by collaborating with the various medical professionals in the hospital. Previous studies in rural areas have demonstrated a professional hierarchy with physicians above other medical professionals in IPC [73,74]. Other medical professionals could face difficulty enforcing their opinions concerning patient care. In addition, rural physicians could be reluctant to participate in IPC based on the beliefs among medical professionals in rural areas [74,75,76]. This research has embarked on solutions for these challenges. Family physicians in the hospital actively communicated with various medical professionals and collected varied opinions and suggestions concerning their patients, leading to better patient care in rural areas and reduced readmission rates.

This study had certain limitations. First, it was performed in a single rural community hospital in Japan, which may have affected the external validity. Family medicine education officially started in Japan 10 years ago; therefore, the educational conditions may not be sufficiently established and different from other countries [77]. The competency of family physicians may be different compared to other countries [78,79]. Future studies should investigate the outcomes for older patients in different types of hospitals in other countries. Second, follow-up could not be maintained in some circumstances. Some patients were discharged to other cities and could not be followed up, which affected the reliability of our findings. Despite these limitations, our findings provide data that may also reflect settings in Japan and other developed countries. The data may be used as a basis for developing relevant guidelines on IPC for better care of older patients to prevent readmission.

## 5. Conclusions

Rural IPC interventions can improve inpatient care of older patients and decrease readmission rates. Comprehensive IPC interventions may require the collaboration of family medicine physicians with dentists, therapists, nutritionists, nurses, and pharmacists in the medical care of patients. For effective IPC interventions, family physicians in hospitals require proactive collaborations with various medical professionals to improve older patients’ care in rural hospitals.

## Figures and Tables

**Figure 1 healthcare-11-00269-f001:**
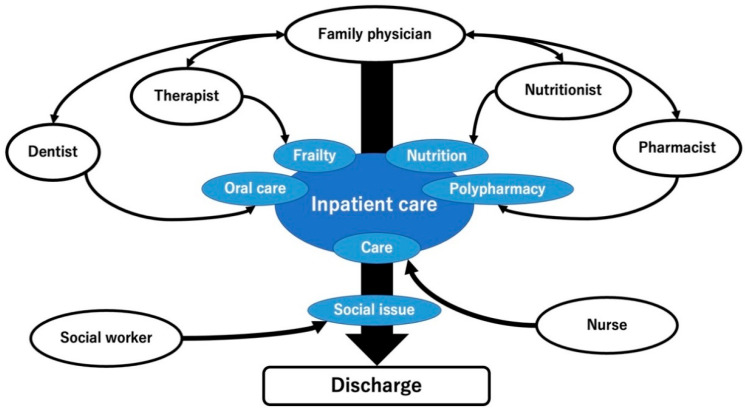
The framework of rural IPC in Unnan City Hospital. IPC, interprofessional collaboration.

**Figure 2 healthcare-11-00269-f002:**
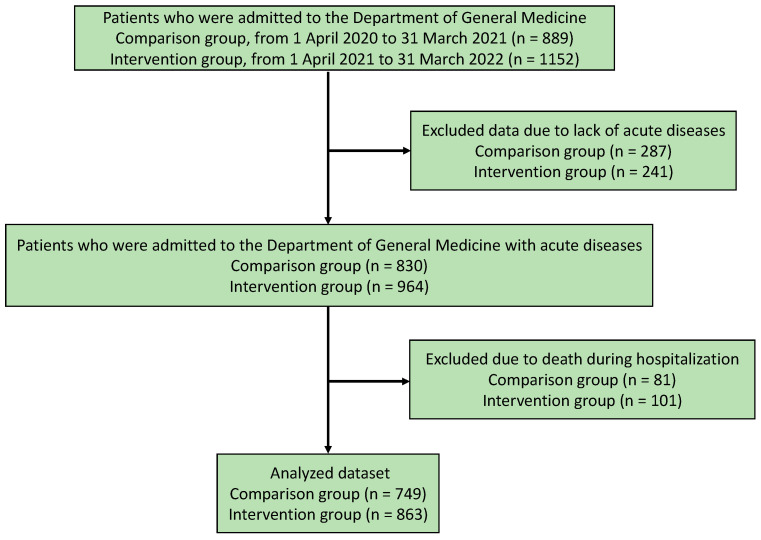
Flowchart of the participant selection.

**Figure 3 healthcare-11-00269-f003:**
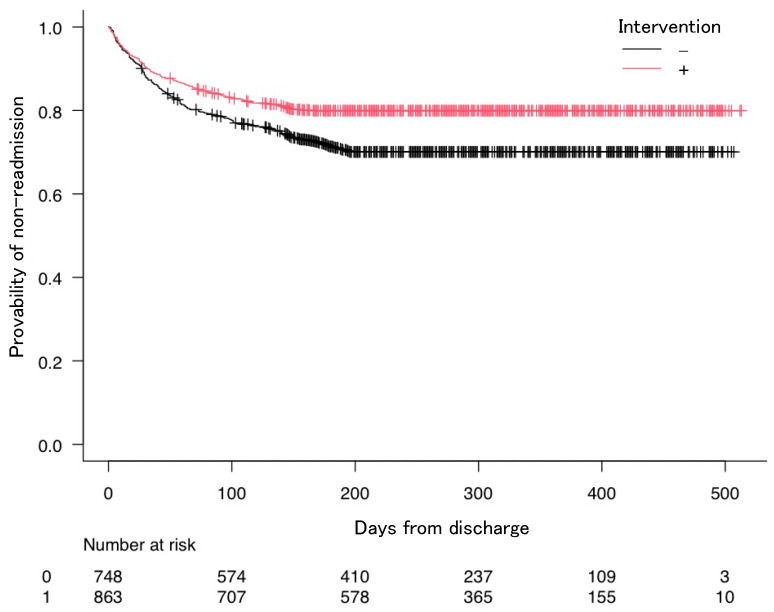
Kaplan–Meier curves showing the probability of non-readmission following discharge. Graphs are presented for the interprofessional collaboration intervention.

**Table 1 healthcare-11-00269-t001:** Concrete interprofessional collaboration of family physicians with other healthcare professionals.

Healthcare Professionals	Purpose	Collaboration
Dentists	To improve the oral care of older patients	Family medicine physicians assessed the admitted patients and discussed the need for oral care with patients and their families. Family physicians informed the dentist and dental hygienists about the reasons for the admission of patients. Dentists intensively assessed the condition of the teeth, gingiva, and mucosa of the mouth.
Physical and occupational therapists	To improve conditions of frailty and adjust effective rehabilitation for each patient	Family physicians and therapists discussed their patients from each other’s perspectives and shared ideas for the care and discharge plan of hospitalized patients once a week. They adjusted the rehabilitation methods and goals.
Pharmacists	To reduce the negative effects of polypharmacy	Family physicians and pharmacists discussed medications of admitted patients to identify unnecessary medications for the patients.
Nutritionists	To improve nutritional conditions	Nutritionists screened the nutritional condition of admitted patients and discussed with family physicians alterations of patient meals upon observation of patient food intake and changes in the nutritional condition based on laboratory nutritional assessment.
Nurses and social workers	To mitigate the difficulty of patients’ lives post-discharge	Weekly discussions were performed regarding patient care and discharge management in each ward among family physicians and chief nurses. The contents of the discussion were shared with social workers. Social workers considered ways to mitigate the difficulty of patient lives post-discharge

**Table 2 healthcare-11-00269-t002:** Demographics of the participants.

Factor	Total	Intervention Group	Comparison Group	*p*-Value
N	1612	863	749	
Age (years), mean (SD)	79.86 (15.38)	79.16 (16.19)	80.68 (14.34)	0.047
Male sex (%)	726 (45.0)	399 (46.2)	327 (43.7)	0.316
Serum albumin (g/dL), mean (SD)	3.70 (2.35)	3.76 (2.79)	3.63 (1.71)	0.266
Hemoglobin (g/dL), mean (SD)	12.23 (7.47)	12.31 (8.33)	12.15 (6.33)	0.680
eGFR (mL/min/1.73 m^2^)	58.93 (22.47)	58.83 (22.93)	59.04 (21.94)	0.849
BMI (kg/m^2^), mean (SD)	21.63 (33.03)	22.27 (45.01)	20.90 (4.05)	0.405
Medicines taken, median, (IQR)	6.00 (0.00, 19.00)	6.00 (0.00, 19.00)	6.00 (0.00, 19.00)	0.420
Patients with polypharmacy, n (%)	1035 (64.2)	557 (64.5)	478 (63.8)	0.795
FIM score at admission				
Motor domain score, median, (IQR)	63.00 (3.00, 91.00)	61.00 (13.00, 91.00)	65.00 (3.00, 91.00)	0.932
Cognitive domain score, median, (IQR)	32.00 (0.00, 64.00)	32.00 (0.00, 64.00)	31.00(5.00, 43.00)	0.610
Readmission (%)	391 (24.3)	172 (19.9)	219 (29.2)	<0.001
Care level (%)				
0	975 (60.5)	518 (60.0)	457 (61.1)	
1	116 (7.2)	61 (7.1)	55 (7.4)	
2	151 (9.4)	79 (9.2)	72 (9.6)	
3	147 (9.1)	79 (9.2)	68 (9.1)	
4	123 (7.6)	67 (7.8)	56 (7.5)	
5	99 (6.1)	59 (6.8)	40 (5.3)	
Dependent condition (%)	636 (39.5)	345 (40.0)	291 (38.9)	0.683
CCI score (%)				
0	90 (5.6)	59 (6.8)	31 (4.1)	
1	42 (2.6)	20 (2.3)	22 (2.9)	
2	81 (5.0)	40 (4.6)	41 (5.5)	
3	135 (8.4)	79 (9.2)	56 (7.5)	
4	325 (20.2)	157 (18.2)	168 (22.4)	
5	315 (19.5)	162 (18.8)	153 (20.4)	
6	283 (17.6)	173 (20.0)	110 (14.7)	
7	181 (11.2)	83 (9.6)	98 (13.1)	
8	85 (5.3)	51 (5.9)	34 (4.5)	
9	51 (3.2)	26 (3.0)	25 (3.3)	
10	16 (1.0)	11 (1.3)	5 (0.7)	
11	4 (0.2)	2 (0.2)	2 (0.3)	
12	2 (0.1)	0 (0.0)	2 (0.3)	
13	1 (0.1)	0 (0.0)	1 (0.1)	
15	1 (0.1)	0 (0.0)	1 (0.1)	
CCI score ≥5 (%)	939 (58.3)	508 (58.9)	431 (57.5)	0.613
Heart failure (%)	304 (18.9)	180 (20.9)	124 (16.6)	
Myocardial infarction (%)	133 (8.3)	81 (9.4)	52 (6.9)	
Asthma (%)	74 (4.6)	37 (4.3)	37 (4.9)	
Kidney diseases (%)	145 (9.0)	85 (9.8)	60 (8.0)	
Peptic ulcer (%)	107 (6.6)	40 (4.6)	67 (9.0)	
Liver diseases (%)	70 (4.3)	41 (4.8)	29 (3.9)	
COPD (%)	89 (5.5)	51 (5.9)	38 (5.1)	
DM (%)	254 (15.8)	144 (16.7)	110 (14.7)	
Brain hemorrhage (%)	125 (7.8)	75 (8.7)	50 (6.7)	
Brain infarction (%)	284 (17.6)	150 (17.4)	134 (17.9)	
Hemiplegia (%)	25 (1.6)	6 (0.7)	19 (2.5)	
Dementia (%)	263 (16.3)	137 (15.9)	126 (16.8)	
Connective tissue diseases (%)	74 (4.6)	45 (5.2)	29 (3.9)	
Cancer (%)	293 (18.2)	148 (17.2)	145 (19.4)	

BMI, body mass index; CCI, Charlson Comorbidity Index; SD, standard deviation; COPD, chronic obstructive pulmonary disease; DM, diabetes mellitus; FIM, functional independence measure; IQR, interquartile range; eGFR, estimated glomerular filtration rate.

**Table 3 healthcare-11-00269-t003:** Results of the Cox hazards model based on the relationship between the readmission rates and intervention of IPC.

Factor	Hazard Ratio	95% CI	*p*-Value
Presence of intervention	0.66	0.54–0.81	<0.001
Age	1	0.99–1.01	0.74
Male sex	1.28	1.04–1.58	0.019
Serum albumin	1.15	1.01–1.31	0.031
BMI	0.94	0.92–0.97	<0.001
Hemoglobin	0.96	0.92–1.00	0.056
Polypharmacy	1.56	1.22–2.00	<0.001
FIM score at admission			
Motor domain score	1	1.00–1.01	0.24
Cognitive domain score	1.01	0.99–1.02	0.21
Dependent condition	1.34	1.03–1.76	0.031
CCI score ≥5 (%)	1.97	1.50–2.59	<0.001

BMI, body mass index; CCI, Charlson comorbidity index; CI, confidence interval; FIM, functional independence measure; IPC, interprofessional collaboration.

## Data Availability

The data presented in this study are available upon request from the corresponding author.

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
