# Peer review of "The Effectiveness of Family Medicine-Driven Interprofessional Collaboration on the Readmission Rate of Older Patients"

_healthcare, 2023, doi:10.3390/healthcare11020269_

Round 1

Reviewer 1 Report

The authors addressed all my comments.

Author Response

Responses to Reviewer 1 Comments:

The authors addressed all my comments.

Response: Thank you for your valuable comments and suggestions

Responses to the Comments of the Academic Editor:

One of the reviewers has suggested that your manuscript should undergo extensive English revisions. Note that extensive English editing is not included in the APC. This may not accurately reflect the English level of your manuscript, but we recommend that you further check the English during revision. If the reviewer does not provide detailed comments, you may ask the assistant editor to contact the reviewers for more specific comments.

Response: Thank you for your valuable comment. We have reviewed the entire manuscript and ensured that there are no English language errors present. Furthermore, the manuscript has been reviewed by a specialized English editing company.

One of reviewers has occurred to us that an unusually high proportion of references in the paper are made to previous work by the authors. To ensure some balance in accordance with standard academic practice, please consider removing some of the references made to your own previous work or adding to the references made to other scholars' publications.

Response: Thank you for your valuable comment. As per your suggestion, we have rechecked all the references and reduced the percentage of self-cited papers to less than 20%.

Reviewer 2 Report

The topic of the article is interesting and important in general for the work of a multi-professional team, which has already been proven in the past to have a significant contribution to the quality of care in health systems.

Some comments:

1. The primary outcome variable of the study was the readmission rate, which is a significant issue in the health care system. Despite its importance, these topic is almost never mentioned in the introduction. It would have been appropriate to briefly present this issue of readmission and its' consequences for both the elderly patient and the health system.

2. Similarly, there is not enough presentation of the subject of preparation for discharge, which is also very important in preventing or reducing re-hospitalizations.

3. On page 8, line 314, the significant differences between the groups are shown when I did not see it (the P values) in table #1.

Author Response

Responses to Reviewer 2 Comments:

The topic of the article is interesting and important in general for the work of a multi-professional team, which has already been proven in the past to have a significant contribution to the quality of care in health systems.

Some comments:

  1. The primary outcome variable of the study was the readmission rate, which is a significant issue in the health care system. Despite its importance, these topic is almost never mentioned in the introduction. It would have been appropriate to briefly present this issue of readmission and its' consequences for both the elderly patient and the health system.

Response: Thank you for your valuable comment. We agree with your point of view. As per your suggestion, we have included a paragraph in the Introduction section emphasizing the significance of readmission rates in healthcare.

“One of the factors that highlights the improvement in hospitalized older patient care is the decrease in readmission rate. Older patients have various chronic diseases with polypharmacy [29,30]. The nature of this multimorbidity can make older patients vulnerable to the exacerbations of chronic diseases, infections, and cardiovascular diseases [31]. Readmission of older patients increases the likelihood of morbidity and mortality, which should be prevented [32,33]. Through effective IPC of various healthcare professionals, the readmission rate can be decreased by enhancing the discharge and follow-up processes in outpatient departments. High readmission rates increase the burden on healthcare systems. They led to increased patient numbers in hospitals, which require the commitment of various medical professionals [34]. The commitment in hospitals is typically more intensive than in that in the community [35]. The increase in hospitalized patient numbers can exhaust healthcare professionals, which may ultimately lead to burnout [36–38]. In addition, the cost of hospital admission is higher than community care, impinging on the financial conditions of heath care systems [39]. Utilizing IPC intervention to decrease readmission rates of older patients can mitigate the burden on healthcare professionals and systems [40].” (Lines 76–92)

  1. Similarly, there is not enough presentation of the subject of preparation for discharge, which is also very important in preventing or reducing re-hospitalizations.

Response: Thank you for your valuable comment. We agree with your point of view. As per your suggestion, we have included a paragraph in the Introduction section that discusses how to prevent and reduce readmission.

“One of the factors that highlights the improvement in hospitalized older patient care is the decrease in readmission rate. Older patients have various chronic diseases with polypharmacy [29,30]. The nature of this multimorbidity can make older patients vulnerable to the exacerbations of chronic diseases, infections, and cardiovascular diseases [31]. Readmission of older patients increases the likelihood of morbidity and mortality, which should be prevented [32,33]. Through effective IPC of various healthcare professionals, the readmission rate can be decreased by enhancing the discharge and follow-up processes in outpatient departments. High readmission rates increase the burden on healthcare systems. They led to increased patient numbers in hospitals, which require the commitment of various medical professionals [34]. The commitment in hospitals is typically more intensive than in that in the community [35]. The increase in hospitalized patient numbers can exhaust healthcare professionals, which may ultimately lead to burnout [36–38]. In addition, the cost of hospital admission is higher than community care, impinging on the financial conditions of heath care systems [39]. Utilizing IPC intervention to decrease readmission rates of older patients can mitigate the burden on healthcare professionals and systems [40].” (Lines 76–92)

  1. On page 8, line 314, the significant differences between the groups are shown when I did not see it (the P values) in table #1.

Response: Thank you for your valuable comment. As per your suggestion, we have revised the recommended portions in Table 1.

Responses to the Comments of the Academic Editor:

One of the reviewers has suggested that your manuscript should undergo extensive English revisions. Note that extensive English editing is not included in the APC. This may not accurately reflect the English level of your manuscript, but we recommend that you further check the English during revision. If the reviewer does not provide detailed comments, you may ask the assistant editor to contact the reviewers for more specific comments.

Response: Thank you for your valuable comment. We have reviewed the entire manuscript and ensured that there are no English language errors present. Furthermore, the manuscript has been reviewed by a specialized English editing company.

One of reviewers has occurred to us that an unusually high proportion of references in the paper are made to previous work by the authors. To ensure some balance in accordance with standard academic practice, please consider removing some of the references made to your own previous work or adding to the references made to other scholars' publications.

Response: Thank you for your valuable comment. As per your suggestion, we have rechecked all the references and reduced the percentage of self-cited papers to less than 20%.

Round 2

Reviewer 2 Report

The article is much better.

Good Luck

Author Response

Response:
Thank you for reviewing my article.